# How Are Physical Activity and Mindfulness Associated with Psychological Symptoms Among Chinese University Students: The Independent and Joint Role

**DOI:** 10.3390/bs14111088

**Published:** 2024-11-13

**Authors:** Xiaoqi Wei, Xinli Chi, Sitong Chen, Kaixin Liang, Yue Zhao, Sha Xie

**Affiliations:** 1School of Psychology, Shenzhen University, Shenzhen 518061, China; xiaoqiwei@szu.edu.cn (X.W.); xinlichi@szu.edu.cn (X.C.); 2The Shenzhen Humanities & Social Sciences Key Research Bases of the Center for Mental Health, Shenzhen University, Shenzhen 518061, China; 3Institute for Health and Sport, Victoria University, Melbourne 8001, Australia; sitong.chen@live.vu.edu.au; 4Department of Psychology, Faculty of Social Sciences, University of Macau, Macau, China; yc37309@um.edu.mo; 5Faculty of Education, University of Macau, Macau, China; yc47116@um.edu.mo; 6Faculty of Education, Shenzhen University, Shenzhen 518061, China

**Keywords:** physical activity, mindfulness, depression, anxiety, internet addiction

## Abstract

Prevalence of psychological symptoms among Chinese university students is relatively high globally, and most students cannot receive timely treatment. Therefore, exploring protective factors for these symptoms is crucial. In this study, the aim was to examine the independent and joint associations of physical activity (PA) and mindfulness with symptoms of depression, anxiety, and internet addiction. Due to the simplicity of implementation in a university context, a cross-sectional survey was conducted in China in 2022. Participants were recruited through online advertisements, in which 710 Chinese university students met the eligibility criteria and were included in the analysis. Participants were then classified into four groups according to their PA and mindfulness levels. Adjusted nominal logistic regression models controlled for age and sex were fitted to examine the independent and joint associations of PA and mindfulness with symptoms of depression, anxiety, and internet addiction. When examined independently, high PA was associated with lower odds of depressive, anxiety, and internet addiction symptoms, while high mindfulness was associated with lower odds of depressive symptoms. When examining the joint effect, compared to students with low PA and low mindfulness, those with high PA and low mindfulness had a lower risk of depressive and internet addiction symptoms, while students with high PA and high mindfulness reported the lowest odds of depressive, anxiety, and internet addiction symptoms. The current study found that a combination of PA and mindfulness was associated with a lower risk of psychological symptoms. Future studies are suggested to confirm the joint effects of PA and mindfulness on mental health using experiment design.

## 1. Introduction

From the perspective of life-span development [1], university is a crucial period when students are transitioning from adolescence to adulthood. At this stage, although the physiological development of university students has been relatively mature, the psychological and emotional development is comparatively immature [2]. A wide range of studies demonstrated that a significant proportion of mental disorders have their onset in the adolescence or youth period (12–24 years), with about 75% of lifetime cases emerging by age 25 [3]. Among various mental problems, symptoms of depression, anxiety, and internet in university students are increasingly becoming a public health concern. The global prevalence rates of depressive symptoms varied from 7.9% to 40.1% [4], anxiety symptoms from 7.4% to 55% [5], and internet addiction symptoms from 12.6% to 67.5% [6]. In China, the prevalence rates of depressive and anxiety symptoms among university students were 35.7% and 24.2%, respectively [7,8], which is relatively high globally. The risk of internet addiction symptoms in China was also as high as 36.7% [9]. Meanwhile, these psychological symptoms can also be co-morbid [10] and may further lead to detrimental outcomes, such as impairment of executive function [11], poor interpersonal relationships [12], and academic underperformance [13]. However, only 25.3–36.3% of students with mental health problems have received timely treatment in Western countries [14], and the proportion in China is even smaller [15], leading to the relatively high prevalence of psychological symptoms. Therefore, it is essential to reveal the common protective factors for these three symptoms so that effective and economical intervention programs can be designed for the population at risk.

According to the biopsychosocial model [16], the occurrence of symptoms stems from the interaction of biological, psychological, and social aspects, rather than being influenced by only one aspect. For example, the incidence of depressive symptoms, anxiety symptoms, and internet addiction symptoms are associated with dysregulation of the autonomic nervous system, emotional dysregulation, and perceived inadequate social support. This suggests that researchers should consider finding the protective factors against the development of psychological symptoms from these three aspects simultaneously in order to identify more comprehensive intervention targets, which have been overlooked in most previous studies. It has been well-established that physical activity (PA) and mindfulness levels are protective factors for depressive symptoms, anxiety symptoms, and internet addiction symptoms. Such two protective factors can directly have positive effects on the biological (e.g., regulating autonomic nervous system) [17] and psychological aspects (e.g., improving emotional regulation), respectively [18,19], and can indirectly contribute to improvements in the social aspect (e.g., less social impairment) [18,20]. In addition, given the high time and labor costs of traditional counseling and the side effects of medication on students’ physical health, exercise-based and mindfulness-based interventions were considered two alternative non-pharmacological and promising interventions for the symptoms of depression, anxiety, and internet addiction [21,22]. PA and mindfulness levels are two crucial indicators in understanding and developing these interventions. Therefore, the current study focuses on exploring the associations between these two factors and depressive, anxiety and internet addiction symptoms.

Specifically, PA is defined as any physical movement that requires energy expenditure by skeletal muscles. The amount of physical activity can increase positive emotional arousal and satisfy individual needs (e.g., emotional needs), via activating the sympathetic nervous system to secrete hormones (e.g., dopamine and endorphins) [23]. The increase in positive emotional arousal can help reduce the experience of depressed and anxiety mood. Previous meta-analysis studies found that individuals with high PA had 12–32% and 15–34% lower odds of depressive and anxiety symptoms than those with low levels [24]. Along with individual needs being satisfied, the risk of internet addiction symptoms would decrease [25]. Regarding mindfulness, it refers to an individual’s characteristic tendency to maintain awareness of the present moment in a non-reactive and non-judgmental manner [26]. Mindful individuals were evidenced to have good emotion regulation and resilience [27]. They could observe their present-moment negative thoughts and emotions in a less judgmental manner, which is conducive to the acceptance of unpleasant experiences. Conversely, individuals with lower levels of mindfulness were prone to have irrational beliefs towards stress [28], leading to an increase in the risk of emerging depressive and anxiety symptoms. At the same time, mindfulness levels are also negatively related to internet addiction symptoms [29]. One possible reason is that highly mindful individuals could aware their grasping of positive experiences (e.g., craving) and use adaptive ways to control themselves not being addicted to the internet [30].

The above informed literature demonstrated that physical activity and mindfulness were associated with mental health, suggesting the independent role of physical activity and mindfulness against these symptoms. Therefore, in the present study, it was assumed that high physical activity (Hypothesis 1a) and high mindfulness (Hypothesis 1b) were negatively related to depressive, anxiety, and internet addiction symptoms among Chinese university students. These hypotheses were re-validated as preconditions for further exploring the complex relationship between physical activity and mindfulness with these symptoms.

The biopsychosocial model also highlights the importance of revealing the joint associations between various aspects of protective factors and symptoms, which is crucial for developing comprehensive and effective intervention protocols. In addition, the joint effects of PA and mindfulness levels also provide a holistic perspective of combining psychology and exercise science to explore protective factors of psychological symptoms. However, most studies have only focused on the independent role of PA and mindfulness levels in depressive, anxiety, and internet addiction symptoms, but their joint associations were rarely considered. In fact, there may be a bidirectional relationship between levels of PA and mindfulness. Specifically, individuals with high PA are more likely to develop a high level of mindfulness [31], and high mindfulness people tend to live healthier lives and be more willing to engage in physical activities [32]. Based on the conservation of resources theory [33], the reciprocal relationship between PA and mindfulness confers benefits to the accumulation of positive mental resources that help individuals to better cope with stress, leading to a lower risk of depressive, anxiety and internet addiction symptoms. A recent empirical study showed that individuals with simultaneously high PA and high self-compassion levels (mindfulness as one of the core dimensions) were at a lower risk of depressive symptoms than those with either a single high PA or high self-compassion [34]. From this viewpoint, high PA and high mindfulness levels may have similar joint effects on psychological symptoms. Therefore, we assumed that a combination of high PA and high mindfulness was associated with a lower risk of depressive, anxiety, and internet addiction symptoms among Chinese university students (Hypothesis 2).

Given the fact that both PA and mindfulness levels are malleable factors that can be cultivated and enhanced, integrating them within intervention protocols for mental health promotion is considerable. Therefore, as the first step in determining the potential relevance, the aim of this study was to explore the independent and joint role of PA and mindfulness in the occurrence of depressive, anxiety, and internet addiction symptoms among Chinese university students. This would fill the literature gap of the combined effects of PA and mindfulness, and further provide theoretical and practical implications for future interventions.

## 2. Methods

### 2.1. Participants and Procedures

This cross-sectional study was conducted via an online survey in November 2022. Given the ease of availability and resource restrictions, convenience sampling was used to recruit university students as our study participants through online advertisements which explained the procedure and rewards of this study to participants. Online informed consent was obtained from all participants before completing demographic information (i.e., sex and age) and the online questionnaires, and each participant received RMB 12 (USD = 1.78) as remuneration if they had completed all questionnaires and complied with inclusion criteria. The inclusion criteria were as follows: (a) Chinese university students and non-psychology major, as students majoring in psychology are more familiar with the purpose and procedure of psychological research, which may affect the authenticity of their responses; (b) answer for more than 300 s; (c) passing the attention test through providing right answer to specific items (e.g., “please choose ‘never’ in this item” and were considered as wrongly responding to the item if they chose other options). The latter two inclusion criteria were to ensure that participants read and understood each item carefully to reduce bias in our results. In total, 861 participants initially enrolled in this study, and 151 participants were excluded due to non-compliance with the inclusion criteria, resulting in a final sample of 710 valid response. This sample size exceeded the minimum requirement of 675, determined through a power analysis using G*Power 3.1, which was based on following assumptions: an expected prevalence (26.0%) of symptoms, 1.6 odds ratio (OR) of symptoms among low PA, margin of error of 5%, and power of 80% [35,36].

Information that directly identified an individual (e.g., name, phone number) was removed from the collected data or anonymously coded. Data collected from participants was processed to eliminate any direct personal identifiers, such as full names or phone numbers, to ensure the anonymity of participants. Therefore, all authors had no access to information that could identify participants. This study was conducted in accordance with the Declaration of Helsinki and approved by the Ethics Committee of the College of Medicine, Shenzhen University (2020005, 12 March 2020).

### 2.2. Measures

#### 2.2.1. Physical Activity (PA)

Individuals’ physical activity (PA) was measured by the Chinese version of International Physical Activity Questionnaire Short Form (IPAQ-SF) [37], which has been confirmed as adequately reliable and valid in previous studies. The self-report scale measures the frequency and duration of participants’ PA of different intensities (vigorous-intensity PA, moderate-intensity PA, and low-intensity PA) in the past 7 days. Items included “During the last seven days, how many days did you spend doing vigorous physical activity, such as heavy lifting, toiling, aerobics, or fast cycling?”. Types of PA involve sports, walking, sitting, and activities performed at work or in spare time, etc. PA levels can be indicated in terms of metabolic equivalents of task (MET). One unit of MET is the energy expended when an individual is in a resting state. In the current study, participants were classified as one of two levels, either low-level PA (LPA; less than 600 (METs) min/week) or high-level PA (HPA; more than 600 (METs) min/week).

#### 2.2.2. Mindfulness

The level of mindfulness was measured by the Five Facets Mindfulness Questionnaire (FFMQ) [38]. The scale is a 39-item 5-point Likert scale rated from 1 (never true) to 5 (very often true). It includes five dimensions (non-judging, describing, non-reacting, acting with awareness, and observing) that were summed ranging from 39 to 195, with higher total scores suggesting higher levels of trait mindfulness. Cronbach’s α was 0.83 in our sample. The confirmatory factor analysis (CFA) showed acceptable model fits (χ^2^ = 1205.40, *df* = 149, *p* < 0.001, CFI = 0.91, TLI = 0.90, RMSEA = 0.03, and SRMR = 0.05), suggesting adequate structural validity. According to the medium values (105) in the current sample, participants were categorized into one of two groups, either low-level mindfulness (LM) or high-level mindfulness (HM).

#### 2.2.3. Depressive Symptoms

Depressive symptoms were assessed by the Chinese version of the Beck Depression Inventory-II (BDI-II) [39], which was a reliable measure in the previous study (Cronbach’s α = 0.89). The scale consists of 21 items scored ranging from 0 to 3 for each item with a total score ranged from 0 to 63. The higher points indicate an increased severity in individual depressive symptoms in the past two weeks. Total scores of 14, 20, and 29 were identified as the cut-off of mild, moderate, and severe symptoms of depression, respectively [40]. The purpose of this study was to examine the odds of occurrence of symptoms in different circumstances, without exploring the odds of symptoms with different severities (the selection of cut-off points for anxiety and internet addiction symptoms also followed this). Thus, a total score of 14 was used in this study as a cut-off point to categorize participants into one of two groups, either no depressive symptoms (BDI-II score < 14) or depressive symptoms (BDI-II score ≥ 14). Cronbach’s α coefficient of BDI-II in the current study was 0.96. Good model fit indices of CFA indicated good structural validity of BDI-II (χ^2^ = 848.10, *df* = 63, *p* < 0.001, CFI = 0.93, TLI = 0.92, RMSEA = 0.07, and SRMR = 0.04).

#### 2.2.4. Anxiety Symptoms

Anxiety symptoms were assessed by the Chinese version of the Generalized Anxiety Disorder scale (GAD-7) [41,42]. The scale consists of seven items with four response options from 0 (not at all) to 3 (nearly every day) that were summed to obtain total scores ranging from 0 to 21 (a higher total score indicating more severe anxiety). A total score greater than or equal to 10 and 15 were identified as mild and severe anxiety, respectively. Thus, with the cut-off point set at 10 in the current study [41], participants were classified into one of two groups, either no anxiety symptoms (GAD-7 score < 10) or anxiety symptoms (GAD-7 score ≥ 10). In the current study, GAD-7 showed high internal consistency reliability (Cronbach’s α = 0.91), and trustworthy model fit indices of CFA (χ^2^ = 62.93, *df* = 21, *p* < 0.001, CFI = 0.98, TLI = 0.98, RMSEA = 0.07, and SRMR = 0.02), suggesting good structural validity.

#### 2.2.5. Internet Addiction Symptoms

The 20-item Internet Addiction Test (IAT-20) [43] was used to assess internet addiction symptoms. Each item scores from 1 (rarely) to 5 (always), and total scores range from 20 to 100, with higher total scores indicating more severe internet addiction symptoms. A cut-off point of 50, which was widely used in previous studies [44], was used in the current study to categorize participants into one of two groups, either no internet addiction symptoms (IAT-20 < 50) or internet addiction symptoms (IAT-20 ≥ 50). Cronbach’s α coefficient for IAT-20 in this study was 0.92. The CFA indicated acceptable model fits, χ^2^ = 704.86, *df* = 75, *p* < 0.001, CFI = 0.92, TLI = 0.90, RMSEA = 0.07, and SRMR = 0.05, suggesting adequate structural validity.

### 2.3. Data Analysis

Subjects had to fill in all items to successfully submit the online questionnaires, so there were no missing values to deal with. Descriptive analyses were conducted to show participants’ characteristics. Mean with standard deviation and frequency with percentage were presented as the descriptive data for continuous variables and categorical variables, respectively. *T*-tests and chi-square tests were conducted to compare age and sex difference among depressive, anxiety, and internet addiction symptoms. Since the data were collected by self-report measures, the Harman single-factor method was performed to test common method biases before further analysis.

Adjusted nominal logistic regression models controlled for age and sex were built to examine the independent and joint associations between PA and mindfulness with depressive, anxiety, and internet addiction symptoms. In the model that examined the independent associations between PA and mindfulness with the above symptoms, PA and mindfulness were also mutually adjusted for each other (i.e., when PA was modeled as the main exposure, the analysis was adjusted for mindfulness, and when mindfulness was modeled as the main exposure, the analysis was adjusted for PA). In the model that examined the joint associations between PA and mindfulness with the above symptoms, our samples were divided into 4 (2 × 2) groups (i.e., Group I: LPA + LM, Group II: HPA + LM, Group III: LPA + HM, Group IV: HPA + HM). Using LPA and LM as the reference group, the odds ratios (ORs) and 95% confidence intervals (CIs) of symptoms for other groups were presented after adjusting for controlled variables. The larger the OR, the more likely the development of symptoms compared to the reference group, while OR < 1 indicated a lower prevalence of symptoms in a group than in the reference group [45]. If 95% CIs did not contain 1, ORs were considered to be statistically significant at 5% level [45]. Results with *p* < 0.05 were also considered as significant. *R*^2^ values indicated the explanatory power of the model for outcome variables. All statistical analyses were performed using SPSS 26.0.

## 3. Results

Among the 710 participants (*M*_age_ = 20.95 years, *SD* = 1.82; 45.1% males), 21.1% participants were classified to the group with low physical activity while 49.3% participants were considered as low mindfulness. Meanwhile, 25.1%, 19.7%, and 54.8% participants reported as having depressive, anxiety, and internet addiction symptoms in the current study, respectively. There were no significant differences in sex for the above symptoms (*p*_depressive_ = 0.97, *p*_anxiety_ = 0.46, *p*_Internet_ = 0.84). Differences in age were found only in internet addiction symptoms (mean difference = 0.297, *t* = 2.17, *p* = 0.03). The Harman single-factor analysis showed that the variance interpretation rate of the first common factor was 21.40%, less than critical standard 40%, indicating that there was no common method bias.

The results of independent associations between PA and mindfulness with depressive, anxiety, and internet addiction symptoms showed that high PA and a high level of mindfulness were two independent protective factors (see Table 1). Specifically, participants with high PA had less risk than those with low PA to develop depressive (OR = 0.44, 95% CI [0.30, 0.65], *p* < 0.001), anxiety (OR = 0.53, 95% CI [0.34, 0.80], *p* < 0.001), and internet addiction (OR = 0.62, 95% CI [0.42, 0.90], *p* = 0.01) symptoms, after adjusting for age, sex, and level of mindfulness, supporting Hypothesis 1a. Likewise, compared to LM, participants with HM were less likely to report depression symptoms (OR = 0.65, 95% CI [0.46, 0.91], *p* = 0.01), but have no difference on anxiety (OR = 0.89, 95% CI [0.61, 1.30], *p* = 0.55) and internet addiction symptoms (OR = 0.76, 95% CI [0.56, 1.02], *p* = 0.07), partially supporting Hypothesis 1b.

Regarding the joint effects, although the *R*^2^ values were relatively low in each model (*R*^2^ _depressive_ = 0.05, *R*^2^ _anxiety_ = 0.02, *R*^2^ _Internet_ = 0.03), there were significant joint associations between PA and level of mindfulness with depressive, anxiety, and internet addiction symptoms, implying that high PA plus high levels of mindfulness can play a more protective role (see Table 2). Specifically, compared to Group I (LPA + LM), Group Ⅳ (HPA + HM) had significantly lower odds of depressive symptoms (OR = 0.32, 95% CI [0.18, 0.55], *p* < 0.001), anxiety symptoms (OR = 0.51, 95% CI [0.27, 0.88], *p* = 0.03), and internet addiction symptoms (OR = 0.42, 95% CI [0.25, 0.73], *p* < 0.001). Group Ⅱ (HPA + LM) was significantly associated with less risk of depressive (OR = 0.57, 95% CI [0.33, 0.97], *p* = 0.04) and internet addiction symptoms (OR = 0.53, 95% CI [0.31, 0.91], *p* = 0.02) than Group Ⅰ, whereas such association was not found on anxiety symptoms (OR = 0.63, 95% CI [0.35, 1.14], *p* = 0.13). However, there were no significant differences between Group Ⅰ and Group Ⅲ (HM + LPA) among these symptoms (*p*s > 0.05). These findings indicated that participants with HPA and HM had the lowest estimate of these three symptoms, supporting Hypothesis 2.

## 4. Discussion

In the present study, the independent and joint associations were investigated between PA and mindfulness with depressive, anxiety, and internet addiction symptoms among Chinese university students. Our findings showed that high PA was negatively associated with depressive, anxiety, and internet addiction symptoms, while high mindfulness was correlated with lower odds of depressive symptoms. Furthermore, compared to the group with LPA and LM, the group with HPA and LM had lower risks of depressive and internet addiction symptoms. The group with HPA and HM presented the lowest odds of depressive, anxiety, and internet addiction symptoms among the four groups. This suggests that a combination of PA and mindfulness may be more strongly associated with reduced depressive, anxiety, and internet addiction symptoms than PA or mindfulness alone. These findings will be further discussed within the biopsychosocial model and existing literature.

Consistent with previous studies, our results support Hypothesis 1a and show significantly negative associations of higher levels of PA with depressive, anxiety, and internet addiction symptoms [46,47,48]. The importance of PA for the mental health of university students is reconfirmed. These findings may be interpreted that PA can directly influence biological health, which in turn contributes to the improvement in mental health. For instance, research has shown that higher PA could reduce interleukin 6 (a kind of inflammatory factor) and produce enough dopamine (a kind of hormone), which may help to improve cognitive function and emotional regulation, thereby reducing symptoms of depression and anxiety [49,50]. Along with changes in biological indicators and improvement of mental state, healthy behaviors will be further promoted. Students may spend more time on other physically and mentally beneficial activities rather than ruminate and/or addict to the internet. Their perceiving of social support would be enhanced when engaging in healthy activities [51]. Therefore, it is understandable that HPA students in this study were found to be at less risk for depressive, anxiety, and internet addiction symptoms, compared to those with LPA who could not obtain the PA benefits described above.

With respect to mindfulness, higher mindfulness levels were associated with less possibilities of depressive symptoms, which was in line with previous studies [52,53] and partially supported Hypothesis 1b. One of the core elements of mindfulness is acceptance [54]. Individuals with high mindfulness levels may manifest a high level of acceptance. Many studies have shown that higher acceptance was related to lower activation of the amygdala, which contributed to adaptive responses to stress events [55]. In addition, they generally take a curious and open-minded view of every moment, which helps break a negative mindset and release from depressive mood [56]. In terms of the relationship between the mindfulness levels with anxiety and internet addiction symptoms, there are interesting findings that are inconsistent with most previous studies; no significantly different risk of anxiety and internet addiction between high mindfulness and low mindfulness group. This may be due to the fact that depressive symptoms are mainly related to persistent sad moods which are deterministic, whereas anxiety symptoms are mainly related to worries about the future which are more uncertain and influenced by more factors. In addition, the internet has become one of the essential parts of students’ daily life. Even high-mindfulness students may frequently use the internet for study or socialization, thus increasing the opportunities for exposure to the internet, leading to the emergence of internet addiction symptoms. Therefore, for anxiety and internet addiction symptoms, the protective roles of mindfulness may not be able to resist the influence of other external risk factors. Other protective factors are needed to combine, such as physical activity, which is consistent with the biopsychosocial model that suggests a simultaneous focus on protective factors targeting different aspects of symptom causation [57]. Our study on the combined effect of PA and mindfulness on anxiety and internet addiction symptoms also partially explains the above interesting results. We found that there is no significantly different risk on depressive, anxiety, and internet addiction symptoms between students with HM and LPA and those with LM and LPA. This implies that high levels of mindfulness may not alleviate the adverse effect of low PA on mental health.

Notably, as Hypothesis 2 expected, the combination of HPA and HM was associated with the lowest odds of depressive, anxiety, and internet addiction symptoms among four groups. These findings were in line with the conservation of resources theory [33], suggesting that positive resources can mutually reinforce to decrease the risk of symptoms. Although our data cannot fully clarify the underlying mechanisms of such a process, some previous evidence can provide partial explanations. Specifically, on the one hand, individuals with HPA are prone to have a greater body awareness which has also been suggested to be a core component of mindfulness [58]. In addition to high awareness, individuals with high mindfulness may also have a relatively high acceptance of negative sensations [59]. These people may maintain and enjoy exercise even though negative sensations are likely to perceive during exercise (e.g., fatigue) [60], implying less time spent on rumination and internet using. On the other hand, engaging in PA can activate the sympathetic nervous system to increase positive emotional arousal, and a highly mindful individual may experience a more mindful state which is associated with the activation of parasympathetic nervous system [17]. Put differently, individuals with both high PA and high mindfulness levels are more likely to regulate these two systems flexibly, thereby reducing stress-related symptoms. In addition, considering that exercise and mindfulness interventions contribute to the improvement in PA and mindfulness levels, respectively [22,61], previous intervention studies have also indirectly explained our results. Previous intervention studies found that compared to a single intervention, the combination of exercise and mindfulness can better decrease emotional dysregulation and increase self-worth among adults [62,63]. It indicates that students with both HPA and HM may be good at adopting adaptive emotion regulation strategies and feel a greater sense of self-worth, thus reducing the risk of depressive and anxiety symptoms. Likewise, a meta study among university students demonstrated that a combination of exercise and psychological interventions (including mindfulness) was more effective than independent interventions on internet addiction symptoms [64]. Additionally, Mallia et al. (2020) have demonstrated the efficacy of media literacy interventions in enhancing health-related behaviors in sports science students, underscoring the potential benefits of comprehensive approaches that target both the psychological and physical health outcomes [65]. Nevertheless, the literature concerning the joint effect between HPA and HM is still rare, and mechanisms of the combined effect are called for future research.

### Limitations and Implications

The limitations of the current study warrant mention. First, the cross-sectional design does not allow us to infer the causality of any associated factor or underlying mechanisms. Future longitudinal studies are needed to track the changes in PA, mindfulness levels, and psychological symptoms over time, providing robust evidence of the casual relationships between the joint effects of PA and mindfulness with depressive, anxiety, and internet addiction symptoms. Second, each model exploring joint relationships has relatively low interpretable power, indicating that additional unexamined factors may provide mechanisms underlying these joint links. Thus, more predictors could be included, and structural equation models could be used to identify potential mechanisms underlying these associations in further studies. Third, the convenience sampling was used to recruit participants in our study, which may lead to sample bias. Of note, the prevalence of depressive and anxiety symptoms in our study fell within the mid-range of global data, and the prevalence of internet addiction symptoms was notably higher. This discrepancy could reflect the cultural or regional differences in these symptoms, which warrant further investigation. Future studies are recommended using stratified random sampling for subject recruitment to improve the representativeness of the sample or extended to more culturally and geographically diverse samples to improve the generalizability of the current study. Fourth, the assessment of physical activity was self-reported and that may cause bias due to memory errors. Using an accelerometer to assess objective PA is highly recommended to reduce such bias and verify the reliability of our findings in future studies. Fifth, the protective effects of PA and mindfulness levels on psychological symptoms of varying severities may be different, suggesting different dosages of further interventions. Future research is encouraged to examine the joint effects of PA and mindfulness levels on diverse severities of psychological symptoms (i.e., depressive, anxiety, and internet addiction symptoms) to provide more precise guidelines for the development of intervention programs. Lastly, the independent protective effect of mindfulness levels on the three psychological symptoms seems to be different. Future studies should further examine the moderators that might influence the independent protective effects of mindfulness on anxiety and internet addiction symptoms.

Despite the limitation, in the present study, an insightful understanding is provided of the relationship between PA and mindfulness with depressive, anxiety, and internet addiction symptoms among Chinese university students. A significant strength of this study was the innovative combination of PA and mindfulness to explore their joint effects, highlighting the co-existence of HPA and HM contributed to less risk of psychological symptoms (i.e., depressive, anxiety, and internet addiction symptoms). It adds empirical evidence for the biopsychosocial model and conservation of resources theory. The results of the current study also provide practical implications from a holistic perspective of combining psychology and sports science. Specifically, individuals with high levels of physical activity in our study participated in exercise more than 600 (METs) min/week, and previous studies have shown that mindfulness practice can improve mindfulness levels. Therefore, simultaneously engaging in exercise of more than 600 (METs) min/week and increasing mindfulness practice may provide more effective prevention and/or intervention for depressive, anxiety, and internet addiction symptoms. A growing body of research also implied that interventions considering both mental and physical elements were more effective for psychological symptoms [66,67]. Institutions of universities or policymakers can add PA components to mindfulness-based interventions (e.g., mindful running). Future research can assess the change in the scores of mental symptoms to explore whether the effectiveness of the combined intervention is better than a single intervention.

## 5. Conclusions

The current study reconfirmed that PA and mindfulness can independently contribute to a lower risk of depressive, anxiety, and internet addiction symptoms, and further found the combination of these two factors could jointly decrease the odds of these symptoms. These findings highlight the importance of considering PA and mindfulness simultaneously in developing relevant interventions for university students. When designing or implementing psychological interventions, incorporating exercise may enhance the effect. Future studies are suggested to confirm the joint effects of PA and mindfulness on mental health outcomes using more improved studies.

## Figures and Tables

**Table 1 behavsci-14-01088-t001:** Independent associations of physical activity (pa) and level of mindfulness with the odds ratios of depressive, anxiety, and internet addiction symptoms.

		Depressive Symptoms	Anxiety Symptoms	Internet Addiction Symptoms
Age		0.98 (0.89–1.08)	0.94 (0.85–1.05)	0.90 (0.83–0.98) *
Sex	Female	1 (ref)	1 (ref)	1 (ref)
	Male	1.08 (0.76–1.54)	1.24 (0.85–1.80)	1.05 (0.78–1.43)
Physical Activity	Low PA	1 (ref)	1 (ref)	1 (ref)
	High PA	0.44 (0.30–0.65) ***	0.53 (0.34–0.80) ***	0.62 (0.42–0.90) *
Mindfulness	Low Mindfulness	1 (ref)	1 (ref)	1 (ref)
	High Mindfulness	0.65 (0.46–0.91) *	0.89 (0.61–1.30)	0.76 (0.56–1.02)

Notes. Data are presented as OR (95% CI). * *p* < 0.05, *** *p* < 0.001.

**Table 2 behavsci-14-01088-t002:** Joint associations of physical activity (pa) and mindfulness (m) with the odds ratios of depressive, anxiety, and internet addiction symptoms.

	Depressive Symptoms	Anxiety Symptoms	Internet Addiction Symptoms
Group I: LM LPA	1 (ref)	1 (ref)	1 (ref)
Group II: LM HPA	0.57 (0.33–0.97) *	0.63 (0.35–1.14)	0.53 (0.31–0.91) *
Group III: HM LPA	0.95 (0.49–1.84)	1.16 (0.57–2.37)	0.60 (0.31–1.17)
Group IV: HM HPA	0.32 (0.18–0.55) ***	0.51 (0.28–0.92) *	0.42 (0.25–0.73) ***
*R* ^2^	0.05	0.02	0.03

Notes. Data are presented as OR (95% CI). * *p* < 0.05, *** *p* < 0.001. Using Group Ⅰ as the reference, the odds ratios of the three symptoms between Group Ⅰ and other groups were compared after adjusting for age and sex.

## Data Availability

Due to the sensitive nature of participants, the data cannot be made available to the public.

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
