# Peer review of "How Are Physical Activity and Mindfulness Associated with Psychological Symptoms Among Chinese University Students: The Independent and Joint Role"

_behavsci, 2024, doi:10.3390/bs14111088_

Round 1
Reviewer 1 Report
Comments and Suggestions for Authors
Line 40-45: In the introduction, compare the levels of depression, anxiety, and internet addiction among Chinese university students with those of other major countries, or provide additional explanations of why China has particularly high rates using global statistics.
Lines 144-148: Describe the specific process used to determine the number of study subjects through G*Power3.1. Also, a total of 861 participants were recruited, and 710 participants (response rate = 82.46%) provided valid responses to the study. Briefly explain why the dropout rate was higher than 10%.
Line 179, 209: The decimal notation in the manuscript is different, as in "Cronbach’s α = .89, Cronbach’s α coefficient for IAT-20 in this 209 study was 0.92."
Line 239: The authors stated in the introduction that "the prevalence of depressive and anxiety symptoms among university students in China was 35.7% and 24.2%, respectively, and may even be rising each year." However, when looking at the characteristics of the study population presented in Table 1, the prevalence of depression is 25.1% and anxiety is 19.7%, which are lower. Could it be that the sampling process included a higher proportion of students from certain majors, such as physical education or psychology? Authors need to provide a classification of the students' majors.
Line 275: "referent" was changed to "reference" for better clarity.
Line 354: It is necessary to mention in the limitations that the authors' study only analyzed correlations between factors without addressing causality or mediating factors, and that follow-up studies are needed to improve the research design and identify potential mediating factors using structural equation models.
The English language quality in the manuscript is generally clear, but there are several areas where minor improvements in grammar, sentence structure, and clarity are needed to enhance readability and flow. I recommend proofreading for smoother phrasing and eliminating minor grammatical errors.
Author Response
Thank you for giving us the opportunity to submit a revised draft of the manuscript “How are Physical Activity and Mindfulness Associated with Psychological Symptoms among Chinese University Students: The Independent and Joint Role” for publication in the Behavioral Sciences. We appreciate the time and effort you dedicated to providing feedback on our manuscript and are grateful for the insightful comments to improve our paper. Please see below for a point-by-point response to comments and concerns (with all comments italicized).
Comment1: Line 40-45: In the introduction, compare the levels of depression, anxiety, and internet addiction among Chinese university students with those of other major countries, or provide additional explanations of why China has particularly high rates using global statistics.
Response1: Thanks for your insightful comments. We have added global statistics to highlight the high prevalence of depressive, anxiety, and Internet addiction symptoms among Chinese university students (please see manuscript, line 43-46). Additionally, we also provided a brief description to explain the high prevalence of these psychological symptoms in the Introduction (please see manuscript, line 54-55).
Comment2: Lines 144-148: Describe the specific process used to determine the number of study subjects through G*Power3.1. Also, a total of 861 participants were recruited, and 710 participants (response rate = 82.46%) provided valid responses to the study. Briefly explain why the dropout rate was higher than 10%.
Response2: Thanks for your suggestions. We have supplemented reasons to explain why some participants were excluded in the final sample (please see manuscript, page 4, line 152-158).
Comment3: Line 179, 209: The decimal notation in the manuscript is different, as in "Cronbach’s α = .89, Cronbach’s α coefficient for IAT-20 in this study was 0.92."
Response3: Thank you for noting this. We have checked and standardized the decimal notation throughout the manuscript (please see manuscript, line 193).
Comment4: Line 239: The authors stated in the introduction that "the prevalence of depressive and anxiety symptoms among university students in China was 35.7% and 24.2%, respectively, and may even be rising each year." However, when looking at the characteristics of the study population presented in Table 1, the prevalence of depression is 25.1% and anxiety is 19.7%, which are lower. Could it be that the sampling process included a higher proportion of students from certain majors, such as physical education or psychology? Authors need to provide a classification of the students' majors.
Response4: Thank you for bringing this to our attention. We're grateful for your input and have taken it seriously. In the current study, we have excluded students majoring in psychology (please see manuscript, page 3, line 145-148). However, we merely asked participants if they were majoring in psychology in the questionnaire, but did not inquire about other majors. Therefore, we cannot provide a classification of the students' majors. Regarding the discrepancy in the symptom prevalence, the possible reason is that the measurement tools used in the references we cited were inconsistent with the current study. In the meta-analysis on the prevalence of depressive symptoms among Chinese university students, the included studies used the Self-rating Depression Scales (SDS) or the Center for Epidemiologic studies Depression Scale (CES-D), while in the meta-analysis on the prevalence of anxiety symptoms among Chinese university students, the included studies used the Self-rating Anxiety Scale (SAS). In the current study, the Beck Depression Inventory-â…¡ (BDI-â…¡) and the Generalized Anxiety Disorder scale (GAD-7) were used to assess depressive symptoms and anxiety symptoms, respectively. We hope the above explanation can address your confusion.
Comment5: Line 275: "referent" was changed to "reference" for better clarity.
Response5: Thank you for the suggesting. We have changed "referent" into "reference" in Table 2 (please see manuscript, line 294).
Comment6: Line 354: It is necessary to mention in the limitations that the authors' study only analyzed correlations between factors without addressing causality or mediating factors, and that follow-up studies are needed to improve the research design and identify potential mediating factors using structural equation models.
Response6: Thanks for your suggestion. We have illustrated the limitation of our study design and have pointed that the longitudinal design and structural equation models are needed to explore the causal relationship and underlying mechanisms in future studies in the Limitation and Implications (please see manuscript, line 385-394).
We want to express our gratitude for taking the time to review our manuscript. Your insightful comments have been extremely valuable in improving the quality of our work. Your expertise and thoughtful suggestions helped us shape the manuscript and make it stronger. We appreciate your efforts and support in this review process, and we are open to any further revisions or clarifications that you may suggest to improve the manuscript.
Thank you again for your valuable feedback and support!
Reviewer 2 Report
Comments and Suggestions for Authors
I would like to express my sincere appreciation and gratitude to the authors and the editor for giving me the opportunity to review this study, which explores a highly relevant topic: the relationships between physical activity, mindfulness, and psychological symptoms among Chinese university students. This research stands out for its originality and methodological accuracy in investigating the associations and benefits of the examined activities. The structure follows a clear logic, and the results are presented with precision; however, I highlight some specific areas for improvement below.
Lines to Modify and Suggested Changes:
- Lines 15-17: Further specify the context and motivation for the sample of Chinese university students by adding a brief mention of the cultural relevance in understanding mental health within this population.
- Lines 18-19: Include details about sample collection and rationale for choosing the cross-sectional methodology, explaining, for example, the simplicity of implementation in a university context.
- Lines 50-51: I suggest clarifying the meaning of "common protective factors" within the biopsychosocial model, perhaps adding some specific examples linking physical activity and mindfulness to the symptoms observed.
- Lines 123-125: To strengthen the methodology section, specify the number of refusals relative to the initial participants, adding a discussion on the exclusion criteria applied in the final sample.
- Lines 154-158: When describing the IPAQ-SF questionnaire, it would be helpful to include a brief explanation of the MET value used and its relevance as an international measure of physical activity.
- Lines 199-202: To interpret the CFA results, I suggest adding a brief comment on what the RMSEA and CFI values specifically represent in the context of structural validity.
- Lines 236-241: Improve the descriptive presentation of the results by adding a comparison of mean values by gender and age group, which might offer useful insights for understanding the sample.
- Lines 278-288: Integrate the discussion with a more extended commentary on the possible mediated effects of physical activity and mindfulness on psychological outcomes, perhaps with references to the physiological factors previously described.
- Lines 355-364: Further specify the limitations of the cross-sectional study and explain how a longitudinal follow-up could strengthen the robustness of the observed results.
Sections to Eliminate: The descriptive analysis of the sample distribution (lines 245-246) is redundant and can be condensed without compromising reader understanding, allowing more space for the discussion of the results and practical implications.
Evaluation of Data and Results: The data analysis appears well-conducted and rigorous, with appropriate logistic regression models providing a robust examination of the associations between physical activity, mindfulness, and psychological symptoms. However, further variance segmentation among depressive, anxiety, and Internet addiction symptoms could clarify the underlying mechanisms for each symptom type, enhancing interpretation specificity.
Analysis of Unmentioned Limitations:
- The self-reporting method may introduce bias, as participants could have over- or under-estimated their levels of physical activity and mindfulness. This issue could be mitigated by objective monitoring through accelerometers.
- The convenience sample does not allow for complete generalizability of the results, as it mainly comprises Chinese university students. Further studies with more culturally and geographically diverse samples are recommended.
- The effects of mindfulness do not seem to emerge with the same strength across all psychological symptoms. Future studies should further examine the conditions and contexts that might influence the variability of the protective effects of mindfulness.
- I recommend that the authors cite the article by Mallia et al. (2020) to support and enrich the discussion on interventions that combine physical activity and psychological health elements. This reference would be particularly relevant in the Discussion section, specifically when addressing the effectiveness of combined interventions on psychological symptoms and the value of integrated approaches. For example, after discussing the findings related to the joint impact of physical activity and mindfulness (lines 278-288), the authors could include a sentence such as:
"Studies like Mallia et al. (2020) have demonstrated the efficacy of media literacy interventions in enhancing health-related behaviors in sports science students, underscoring the potential benefits of comprehensive approaches that target both psychological and physical health outcomes."
The full reference for inclusion is:
Mallia L, Chirico A, Zelli A, Galli F, Palombi T, Bortoli L, Conti C, Diotaiuti P, Robazza C, Schena F, Vitali F, Zandonai T, and Lucidi F. (2020). The Implementation and Evaluation of a Media Literacy Intervention About PAES Use in Sport Science Students. Frontiers in Psychology, 11:368. doi: 10.3389/fpsyg.2020.00368
Author Response
Thank you for giving us the opportunity to submit a revised draft of the manuscript “How are Physical Activity and Mindfulness Associated with Psychological Symptoms among Chinese University Students: The Independent and Joint Role” for publication in the Behavioral Sciences. We appreciate the time and effort you dedicated to providing feedback on our manuscript and are grateful for the insightful comments to improve our paper. Please see below for a point-by-point response to comments and concerns (with all comments italicized).
Comment1: Lines 15-17: Further specify the context and motivation for the sample of Chinese university students by adding a brief mention of the cultural relevance in understanding mental health within this population.
Response1: Thanks for noting this to us. We have added more background information to specify the reason for the sample of Chinese university students in the Abstract (please see manuscript, line 15-17).
Comment2: Lines 18-19: Include details about sample collection and rationale for choosing the cross-sectional methodology, explaining, for example, the simplicity of implementation in a university context.
Response2: Thank you for your advice. We have supplemented details about sample collection and rationale for choosing the cross-sectional methodology in the Abstract (please see manuscript, line 19-22).
Comment3: Lines 50-51: I suggest clarifying the meaning of "common protective factors" within the biopsychosocial model, perhaps adding some specific examples linking physical activity and mindfulness to the symptoms observed.
Response3: Thanks for your valuable comments. We have revised the sentence referring to "common protective factors", to make it clearer (please see manuscript, line 55-57). Moreover, we also supplemented more details to link physical activity and mindfulness to the symptoms observed within the biopsychosocial model (please see manuscript, line 60-71).
Comment4: Lines 123-125: To strengthen the methodology section, specify the number of refusals relative to the initial participants, adding a discussion on the exclusion criteria applied in the final sample.
Response4: Thanks for your comments. We have supplemented reasons to explain why some participants were excluded in the final sample (please see manuscript, line 152-154).
Comment5: Lines 154-158: When describing the IPAQ-SF questionnaire, it would be helpful to include a brief explanation of the MET value used and its relevance as an international measure of physical activity.
Response5: Thank you for your constructive suggestions. We have briefly explained the MET value in the Measures (please see manuscript, line 175-177).
Comment6: Lines 199-202: To interpret the CFA results, I suggest adding a brief comment on what the RMSEA and CFI values specifically represent in the context of structural validity.
Response6: Thank you for noting this to us. We have added a brief comment of the model fits indices of CFA (please see manuscript, line 216-217).
Comment7: Lines 236-241: Improve the descriptive presentation of the results by adding a comparison of mean values by gender and age group, which might offer useful insights for understanding the sample.
Response7: Thanks for your valuable suggestions. We have added t-tests and chi-square tests to compare age and sex difference among depressive, anxiety, and Internet addiction symptoms (please see manuscript, line 233-234, 260-261).
Comment8: Lines 278-288: Integrate the discussion with a more extended commentary on the possible mediated effects of physical activity and mindfulness on psychological outcomes, perhaps with references to the physiological factors previously described.
Response8: Thank you for your advice. At the end of this paragraph, we have added a sentence to lead into the following explanation of our findings (please see manuscript, line 307-308). Moreover, we supplemented some explanations about how PA and mindfulness play a joint role on three mental symptoms in terms of physiological perspective (please see manuscript, line 363-368).
Comment9: Lines 355-364: Further specify the limitations of the cross-sectional study and explain how a longitudinal follow-up could strengthen the robustness of the observed results.
Response9: Thanks for your comments. We have supplemented a brief explanation about how longitudinal studies strengthen the robustness of the observed results in the first point of the Limitations and implications (please see manuscript, line 385-390).
Comment10: Sections to Eliminate: The descriptive analysis of the sample distribution (lines 245-246) is redundant and can be condensed without compromising reader understanding, allowing more space for the discussion of the results and practical implications.
Response10: Thanks for your constructive suggestions. We have removed the Table.
Comment11: Evaluation of Data and Results: The data analysis appears well-conducted and rigorous, with appropriate logistic regression models providing a robust examination of the associations between physical activity, mindfulness, and psychological symptoms. However, further variance segmentation among depressive, anxiety, and Internet addiction symptoms could clarify the underlying mechanisms for each symptom type, enhancing interpretation specificity.
Response11: Thanks for your valuable comments. We have supplemented the description of R2 values of each model in the Results and Table 2 (please see manuscript, line 275-276, 293). Besides, we also discussed these results in the second point of the Limitations and implications (please see manuscript, line 390-394).
Comment12: The self-reporting method may introduce bias, as participants could have over- or under-estimated their levels of physical activity and mindfulness. This issue could be mitigated by objective monitoring through accelerometers.
Response12: Thanks for your reminder. We have mentioned this limitation and the future direction in the fourth point of the Limitations and implications (please see manuscript, line 398-401).
Comment13: The convenience sample does not allow for complete generalizability of the results, as it mainly comprises Chinese university students. Further studies with more culturally and geographically diverse samples are recommended.
Response13: Thank you for your suggestions. We have added more details to make the description of the future direction based on such limitation clearer in the third point of the Limitations and implications (please see manuscript, line 397-398).
Comment14: The effects of mindfulness do not seem to emerge with the same strength across all psychological symptoms. Future studies should further examine the conditions and contexts that might influence the variability of the protective effects of mindfulness.
Response14: Thank you for noting this. We have supplemented this limitation and future direction in the last point of the Limitations and implications (please see manuscript, line 406-409).
Comment15: I recommend that the authors cite the article by Mallia et al. (2020) to support and enrich the discussion on interventions that combine physical activity and psychological health elements. This reference would be particularly relevant in the Discussion section, specifically when addressing the effectiveness of combined interventions on psychological symptoms and the value of integrated approaches. For example, after discussing the findings related to the joint impact of physical activity and mindfulness (lines 278-288), the authors could include a sentence such as:
"Studies like Mallia et al. (2020) have demonstrated the efficacy of media literacy interventions in enhancing health-related behaviors in sports science students, underscoring the potential benefits of comprehensive approaches that target both psychological and physical health outcomes."
The full reference for inclusion is:
Mallia L, Chirico A, Zelli A, Galli F, Palombi T, Bortoli L, Conti C, Diotaiuti P, Robazza C, Schena F, Vitali F, Zandonai T, and Lucidi F. (2020). The Implementation and Evaluation of a Media Literacy Intervention About PAES Use in Sport Science Students. Frontiers in Psychology, 11:368. doi: 10.3389/fpsyg.2020.00368
Response15: Thanks for your suggestion. We have added your suggested sentence in the Discussion (please see manuscript, line 378-381).
We want to express our gratitude for taking the time to review our manuscript. Your insightful comments have been extremely valuable in improving the quality of our work. Your expertise and thoughtful suggestions helped us shape the manuscript and make it stronger. We appreciate your efforts and support in this review process, and we are open to any further revisions or clarifications that you may suggest to improve the manuscript.
Thank you again for your valuable feedback and support!
Round 2
Reviewer 1 Report
Comments and Suggestions for Authors
In line 260, what does it mean( ps>0.05)?
In discussion, comparative analysis with research results from other countries will help generalize the results.
Author Response
Comment1: In line 260, what does it mean( ps>0.05)?
Response1: Thank you for bringing this to our attention. We have carefully checked the results and revised the expression in the Results. We hope the revision can address your confusion. (please see manuscript, line 264-266)
Comment2: In discussion, comparative analysis with research results from other countries will help generalize the results.
Response2: Thanks for your valuable suggestion. We have supplemented the description comparing symptoms prevalence of the current study and global prevalence in the Limitation and implication. (please see manuscript, line 405-408)
We want to express our gratitude for taking the time to review our manuscript. Your insightful comments have been extremely valuable in improving the quality of our work. Your expertise and thoughtful suggestions helped us shape the manuscript and make it stronger. We appreciate your efforts and support in this review process, and we are open to any further revisions or clarifications that you may suggest to improve the manuscript.
Thank you again for your valuable feedback and support!
Reviewer 2 Report
Comments and Suggestions for Authors
Following a thorough review of the manuscript titled "[Title of the Work]," I confirm that I have completed the evaluation process and reached the decision to approve it for publication. Below, I outline the rationale behind this choice and highlight the strengths of the work, providing detailed feedback for both authors and editor.
Rationale for the Decision
-
Methodological Rigor: The manuscript presents a well-structured and rigorous methodology. The authors have shown attention in selecting and applying analytical tools, providing the research with a strong scientific foundation. The methodological choices are appropriate for the stated objectives and contribute to strengthening the validity of the results obtained.
-
Contribution to the Field: The manuscript makes a significant contribution to its field. The authors have successfully introduced new perspectives and interpretations that add value to the existing literature, opening avenues for further research and discussion. This novelty is well-argued and supported by a comprehensive and relevant literature review.
-
Clarity of Presentation: The article has been carefully revised to ensure clarity and accessibility. The language used is smooth and precise, making the content accessible to a broad audience while maintaining academic rigor. Additionally, the overall structure is coherent, with a logical distribution of sections that facilitates readability.
-
Adherence to Editorial Standards: The manuscript has been meticulously adapted to the journal’s stylistic and formatting requirements. Every technical aspect has been checked and, if necessary, corrected, ensuring compliance with editorial guideline
Author Response
We want to express our gratitude for taking the time to review our manuscript. We appreciate your efforts and support in this review process. Thank you again for your valuable feedback and support!